# Five New Species of the Genus *Hymenogaster* (Hymenogastraceae, Agaricales) from Northern China

**DOI:** 10.3390/jof10040272

**Published:** 2024-04-08

**Authors:** Ting Li, Ning Mao, Haoyu Fu, Yuxin Zhang, Li Fan

**Affiliations:** 1Department of Life Sciences, National Natural History Museum of China, Tianqiaonandajie 126, Beijing 100050, China; m_zhou_m@163.com; 2College of Life Science, Capital Normal University, Xisanhuanbeilu 105, Beijing 100048, China; 2210801013@cnu.edu.cn (N.M.); fhy865001572@outlook.com (H.F.); zhangyuxin7237@163.com (Y.Z.)

**Keywords:** Basidiomycota, hypogeous, phylogeny, taxonomy

## Abstract

In this study, five new species from China, *Hymenogaster latisporus, H. minisporus*, *H. papilliformis*, *H. perisporius,* and *H. variabilis,* are described and illustrated based on morphological and molecular evidence. *Hymenogaster latisporus* was distinguished from other species of the genus by the subglobose, broad ellipsoidal, ovoid basidiospores (average = 13.7 μm × 11.6 μm) with sparse verrucose and ridge-like ornamentation (1–1.2 μm high); *H. minisporus* by the ellipsoidal to broadly ellipsoidal and small basidiospores (average = 11.7 μm × 9.5 μm); *H*. *papilliformis* was characterized by the whitish to cream-colored basidiomes, and broadly fusiform to citriform basidiospores with a pronounced apex (2–3 μm, occasionally up to 4 μm high), papillary, distinct warts and ridges, and pronounced appendix (2–3 μm long); *H. perisporius* by the dirty white to pale yellow basidiomes, broad ellipsoidal to ellipsoidal, and yellow-brown to dark-brown basidiospores with warts and gelatinous perisporium; *H. variabilis* by the peridium with significant changes in thickness (167–351 μm), and broad ellipsoidal to subglobose basidiospores ornamented with sparse warts and ridges. An ITS/LSU-based phylogenetic analysis supported the erection of the five new species. A key for *Hymenogaster* species from northern China is provided.

## 1. Introduction

*Hymenogaster* (Hymenogastraceae, Agaricales), established by Carolo Vittadini based on eight species found in Europe [1], is one of the most species-rich genera of false truffles [2,3,4]. *Hymenogaster citrinus* was designated as the type species [5]. Species within this genus can form ectomycorrhiza with a wide range of tree species, mainly including Betulaceae, Ericaceae, Fagaceae, Myrtaceae, Pinaceae, Salicaceae, and Tiliaceae, and display no significant host specificity [6,7,8,9,10,11,12,13,14]; thus, they can assist host plants in nutrient uptake and play an important role in the conservation, restoration, or rebuilding of ecosystems.

Since Vittadini’s original description of this genus, the reliance on morphological features alone has led to persistent taxonomic errors and confusion [5,15,16,17]. The primary morphological characteristics for species delimitation are the color or discoloration of the basidiomes when fresh, and the features of basidiospores (including color, shape, and ornamentation). Molecular sequencing technology significantly altered our understanding of the species delineation within this genus. For instance, Stielow et al. [2] re-examined the genus using ITS analysis and described two new species; Smith et al. [18] employed multiple gene regions to demonstrate that *H. mcmurphyi* was in fact a sequestrate species of *Xerocomellus* (Boletineae, Boletales), rather than a *Hymenogaster* species as previously believed based solely on morphological data. However, molecular sequence-based studies of Chinese *Hymenogaster* remain scarce.

A total of 32 species and variants of *Hymenogaster* have been reported in China based on morphology [19]. Clearly, it is necessary that the occurrence of these *Hymenogaster* species be re-examined and verified with molecular data.

In this study, we employed a combination of morphological and molecular methods to systematically investigate *Hymenogaster* species collected from Beijing and Shanxi Province in China. This approach led to the identification and description of five new species. Two known species reported in previous studies in China [19], *H. arenarius* and *H. citrinus*, were confirmed with morphological and molecular evidence (Figure 1 and Figure 2).

## 2. Materials and Methods

### 2.1. Sample Collections

Samples were systematically collected over a period of six years in China and subsequently examined. These voucher specimens have been accessioned in the Herbarium of the Biology Department at Capital Normal University (BJTC). Macroscopic characteristics of these specimens were described from both fresh and dried materials. For microscopic analysis, thin sections were prepared from dried specimens by hand. These sections were then immersed in a 3% KOH (*w*/*v*) solution and Melzer’s reagent [20] for detailed study. For the scanning electron microscopy (SEM) analysis, basidiospores were positioned onto double-sided adhesive tape affixed to the SEM stub. Subsequently, these samples were uniformly coated with a platinum–palladium film utilizing an ion sputter-coater (HITACHI E-1010). The coated samples were then examined and documented using a Hitachi S-4800 SEM (Hitachi, Tokyo, Japan).

### 2.2. DNA Extraction, PCR Amplification, Sequencing and Nucleotide Alignment

Dried gleba was ground by shaking for 3 min at 30 Hz (Mixer Mill MM 301, Retsch, Haan, Germany) in a 1.5 mL tube together with one 3 mm diameter tungsten carbide ball, and total genomic DNA was extracted using the modified CTAB method [21]. The internal transcribed spacer (ITS) region of nuclear ribosomal DNA (nrDNA) was amplified using primers ITS1f/ITS4 [21,22]. The 28S large subunit (nrLSU) nrDNA region was amplified using primers LR0R/LR5 [23]. PCRs were performed in 50 µL reactions containing 4 µL of DNA template, 2 µL of each primer (10 µM), and 25 µL 2 × Master Mix [Tiangen Biotech Co., Beijing, China]. Amplification reactions were performed as follows: for the ITS gene, initial denaturation at 95 °C for 4 min, followed by 35 cycles at 95 °C for 30 s, 55 °C for 45 s, 72 °C for 1 min, and a final extension at 72 °C for 10 min; for the nrLSU gene, initial denaturation at 95 °C for 4 min, followed by 35 cycles at 95 °C for 30 s, 55 °C for 60 s, 72 °C for 1 min, and a final extension at 72 °C for 10 min. The PCR products were sent to Beijing Zhongkexilin Biotechnology Co., Ltd. (Beijing, China) for purification and sequencing. Validated sequences are stored in the NCBI database (http://www.ncbi.nlm.nih.gov/) (accessed on 30 April 2023) under the accession numbers provided (Table 1).

### 2.3. Phylogenetic Analysis

The ITS-LSU combined dataset was assembled and aligned utilizing the MAFFT algorithm [24], adhering to default parameters. This alignment was further refined through manual adjustments in Se-Al v2.03a [25], ensuring optimal sequence similarity. Alignments of all datasets used in this study were submitted to TreeBASE (No. 31242). ML and BI analysis were used together to construct phylogenetic tree. Maximum likelihood (ML) analysis was performed with RAxML 8.0.14 [26] employing the GTRGAMMAI substitution model with parameters unlinked. ML bootstrap replicates (1000) were computed in RAxML using a rapid bootstrap analysis and search for the best-scoring ML tree. The ML trees were visualized with TreeView32 [27]. Clades with bootstrap support (BS) ≥ 70% were considered significant [28]. Bayesian inference (BI) was conducted using MrBayes v3.1.2 [29] as an additional method of evaluating branch support. In BI analysis, after selecting the best substitution models (GTR + I + G for all positions) determined by MrModeltest v2.3 [30], two independent runs of four chains were conducted for 1,065,000 Markov chain Monte Carlo (MCMC) generations with the default settings. Average standard deviations of split frequency (ASDSF) values were far less than 0.01 at the end of the generations. Trees were sampled every 100 generations after burn-in (well after convergence), and a 50% majority-rule consensus tree was constructed and visualized with TreeView 32 [27]. Clades with Bayesian posterior probability (PP) ≥ 0.95 were considered significantly supported [31].

## 3. Results

### 3.1. Molecular Phylogenetics

The ITS-LSU combined dataset was compiled to elucidate the phylogenetic position of the new species in this study. This comprehensive dataset comprises 97 sequences from 23 different species, inclusive of 72 newly generated sequences derived from Chinese collections. The length of the aligned dataset was 1465 bp after the exclusion of poorly aligned sites, with 651 bp for ITS and 814 bp for nrLSU. Both Maximum Likelihood (ML) and Bayesian Inference (BI) analyses resulted in similar phylogenetic tree topologies. The tree, as deduced from the ML analysis, is presented here, which showed robust statistical bootstrap support from ML and posterior probability values from BI, confirming the reliability of the findings (Figure 1). Our collections were resolved in seven independent clades with strong statistical bootstrap support, indicating they represented seven distinct species. Two of them represented the known species *H. arenarius* and *H. citrinus*, respectively. The remaining five species are novel to science.

### 3.2. Taxonomy

*Hymenogaster latisporus* L. Fan and T. Li, sp. nov. (Figure 3)

MycoBank: MB852599

Etymology: *latisporus*, referring to the spores with a wider width.

Holotype: China. Shanxi Province, Yuncheng City, Yuanqu County, Lishan Town, Shunwangping, alt. 1744 m, 30 October 2017, YXY144 (BJTC FAN1134).

Description—the basidiome is subglobose to irregular globose, 1.2–1.8 cm in diameter, soft and elastic, earth yellow when fresh, yellow-brown to brown when dry, with a distinct depression at the sterile base. Its surface is smooth and glabrous.

The peridium is 127–246 μm thick, prosenchymatous, interwoven, light yellow-brown, 2–3 μm broad hyphae, and mixed with inflated ellipsoid to subglobose cells of 8–13 μm near to hymenium, pale yellow to nearly hyaline. Gleba light yellow-brown to brown when fresh, loculate, locules irregular, empty, filled with spores at maturity. Hymenium 19–28 μm thick. Hymenial cystidia not seen. The basidia are narrow clavate, two-spored, 28.5–40.5 μm long; the sterigmata are short, 1–2 μm long, and the basidia are collapsed and disappeared at maturity. The basidiospores are subglobose, broad ellipsoidal, ovoid, pale yellow-brown to yellow-brown at maturity, and ornamented with warts and ridges 1–1.2 μm high, with ridges short, irregular, and interwoven, (11.6–)12–15.3(–16) × 10.6–12.6 μm (L_m_ × W_m_ = 13.7 ± 0.8 × 11.6 ± 0.5, *n* = 30), Q = 1.1–1.3 (Q_av_ = 1.2), excluding ornamentations, with gelatinous perisporium, with apex, obtuse, papillary, nearly hyaline, 2.5–3 μm high, appendix evident, 1.5–2.5 μm long.

Habit and habitat: hypogeous, gregarious, under the soil of *Pinus tabulaeformis* Carr., alt. 1744 m, Shanxi, northern China.

Notes: *Hymenogaster latisporus* is characterized by the earth yellow basidiome, large and subglobose, broad ellipsoidal, ovoid basidiospores with sparse verrucose and ridge-like ornamentations. Compared with *H. minisporus* sp. nov (spores 10.8–12.6 × 8.5–10.4 μm), another new species in this study, the former has larger spores (12–15.3(–16) × 10.6–12.6 μm), and with sparse spore ornamentations; compared with *H. variabilis* sp. nov (peridium 167–351 μm thick), this species has a thinner peridium (127–246 μm thick). The ITS-LSU-based phylogeny supported the description of this new species. DNA analysis showed that *H. latisporus* shared less than 97% identity in ITS sequence to other *Hymenogaster* species.

*Hymenogaster minisporus* T. Li & L. Fan, sp. nov. (Figure 4)

MycoBank: MB852601

Etymology: *minisporus*, referring to small basidiospores.

Holotype: China. Beijing, Miyun County, Bulaotun Town, alt. 273 m. 27 July 2020, in soil under *Castanea mollissima* Blume, ZH571 (BJTC FAN1244).

Description—the basidiome is subglobose to globose, 0.5–1.7 cm in diameter, soft and elastic, dirty white when fresh, stained pale brown, with a distinct depression at the sterile base. The surface is smooth and glabrous.

The peridium is 110–288 μm thick, pseudoparenchymatous, composed of elliptic cells of 11–16 × 8–11 μm, light yellow-brown to pale yellow; the outer surface of the peridium locally exhibits a layer of more-or-less parallel interwoven hyphae of 4.8–7.5 μm broad, light yellow-brown. The gleba are light brown when fresh, brown when dry, loculate, with locules irregular or oblong, empty, and filled with spores at maturity. The hymenium is 16–22 μm broad. The hymenial cystidia are clavate, 29–38 µm long, and only present when young, collapsed, and disappeared at maturity. The basidia are cylindrical, not inflate on the apex, two-to-four-spored, mostly two-spored, 23–35 μm, sterigmata short, 2–3 μm long, basidia collapsed and disappeared at maturity. The basidiospores are ellipsoidal to broadly ellipsoidal, occasionally subglobose, yellow-brown at maturity, ornamented with warts and irregular short ridges of 1 μm high, (10–)10.8–12.6(–13.3) × 8.5–10.4(–10.9) μm (L_m_ × W_m_ = 11.7 ± 0.6 × 9.5 ± 0.7, *n* = 30), Q = 1.2–1.3 (Q_av_ = 1.2), excluding ornamentations, without gelatinous perisporium, with a small apex, obtuse, papillary, 1–1.3 μm high, with appendix, 1–1.5 μm long.

Habit and habitat: hypogeous, gregarious, in the soil under *Castanea mollissima*.

Notes: *Hymenogaster minisporus* is characterized by the ellipsoidal to broadly ellipsoidal and small basidiospores. This new species was grouped into a clade with the ‘cryptic species 1’ of *H. niveus* provisionally proposed by Stielow et al. [2] but without statistical support (Figure 1). Morphologically, the latter has white basidiomes that change to red when touched or bruised [2]. ITS-LSU-based phylogeny supports the erection of this new species. DNA analysis showed that *H. minisporus* shared less than 97% identity in the ITS sequence to other *Hymenogaster* species.

*Hymenogaster papilliformis* L. Fan & T. Li, sp. nov. (Figure 5)

MycoBank No: MB852600

Etymology: *papilliformis*, referring to the papillia-shaped apex of basidiospores.

Holotype: China. Shanxi Province, Linfen City, Xi Country, Wulu Mountain National Nature Reserve, 37°32.57′ N, 111°12.10′ E, alt. 1730 m, 26 October 2017, in soil under *Quercus* sp., LT051 (BJTC FAN1074).

Description—the basidiome is subglobose to irregular globose, 0.8–2.5 cm in diameter, soft and elastic, whitish to cream-colored when fresh, with pale yellow to pale taupe spots, yellowish to yellow-brown when dry, with a distinct depression at the sterile base. The surface is smooth and glabrous. 

The peridium is 140–300 μm thick, pseudoparenchymatous, composed of elliptic cells of 14–20 μm in diam, pale yellow to nearly hyaline. The gleba are reddish-brown to brown when fresh, deep brown when dry, and loculate; the locules are rectangle to irregular shape, empty, filled with spores at maturity. The hymenium is 40–70 μm thick. The hymenial cystidia are not seen. The basidia are rare, clavate, one-to-two-spored, mostly two-spored, sterigmata short, 2–4 μm long, basidia collapsed and disappeared at maturity. The basidiospores are broadly fusiform to citriform, yellow-brown at maturity, ornamented with distinct warts and irregular short ridges 1 μm high, ridges interwoven, (13.3–)14–18.6(–19.3) × (10–)10.6–13.6(–15) μm (L_m_ × W_m_ = 15.7 ± 1.2 × 11.7 ± 0.8, *n* = 30), Q = 1.3–1.4 (Q_av_ = 1.34), excluding ornamentations, without gelatinous perisporium, with a pronounced apex, obtuse, papillary, nearly hyaline, 2–3 μm high, occasionally up to 4 μm, appendix very evident, truncate, nearly hyaline, 2–3 μm long.

Habit and habitat: hypogeous, gregarious, in coniferous and broadleaf mixed forest, in the soil under *Betula platyphylla* Sukaczev, *Larix gmelini* (Rupr) Rupr., *Pinus tabulaeformis*, *P. bungeana* Zucc. ex Endl., and *Quercus liaotungensis* Koidz.

Additional specimens examined. China. Beijing, Mentougou District, Baihuashan Mountain, 39°49.42′ N, 115°35.14′ E, alt. 1593 m, 14 October 2016, in soil under *Pinus tabulaeformis*, SXY009 (BJTC FAN655); Yanqing District, Badaling Great Wall, 40°33.10′ N, 115°97.40′ E, alt. 809 m, 16 September 2017, in soil under *Quercus liaotungensis*, HKB170 (BJTC FAN1155), HKB171 (BJTC FAN1156). Shanxi Province, Lvliang City, Jiaocheng Country, Pangquangou, 37°51.37′ N, 111°27.18′ E, alt. 1879 m, 6 September 2017, in soil under *Betula platyphylla*, XYY009 (BJTC FAN807), YXY056 (BJTC FAN819), YXY057 (BJTC FAN820), YXY058 (BJTC FAN821); Linfen City, Pu Country, Gelaozhang, 37°32.57′ N, 111°12.10′ E, alt. 1698 m, 10 September 2017, in soil under *Pinus tabulaeformis*, YXY074 (BJTC FAN888), XYY026 (BJTC FAN902), LT025 (BJTC FAN917), HKB109 (BJTC FAN935); Xi Country, Shenjiagou, 36°36.4′ N, 111°10.34′ E, alt. 1321 m, 10 September 2017, in soil under *Pinus bungeana*, YXY077 (BJTC FAN891), XYY034 (BJTC FAN910), LT034 (BJTC FAN925), LT035 (BJTC FAN926); Pu Country, Megou, 36°36.57′ N, 111°13.10′ E, alt. 1443 m, 11 September 2017, in soil under *Larix gmelini*, HKB117 (BJTC FAN944); Chaoyanggou, 36°34.3′ N, 111°11.55′ E, alt. 1645 m, 11 September 2017, in soil under *Pinus tabulaeformis*, HKB120 (BJTC FAN947), YXY046 (BJTC FAN971); ibid., in soil under mixed forest, LT036 (BJTC FAN954), LT037 (BJTC FAN955), in soil under *Quercus liaotungensis*, LT040 (BJTC FAN958), LT (BJTC FAN959), LT042 (BJTC FAN960), LT043 (BJTC FAN961), XYY039 (BJTC FAN965), XYY043 (BJTC FAN969), YXY088 (BJTC FAN979), X.Y. Yan089 (BJTC FAN980), YXY092 (BJTC FAN983); Megou, 36°36.57′ N, 111°13.10′ E, alt. 1443 m, 11 September 2017, in soil under *Quercus liaotungensis*, YXY081 (BJTC FAN972); Gelaozhang, 36°34.8′ N, 111°11.33′ E, alt. 1770 m, 12 September 2017, in soil under *Quercus liaotungensis*, LT046 (BJTC FAN991), XYY047 (BJTC FAN998), in soil under *Pinus tabulaeformis*, YXY099 (BJTC FAN992), YXY100 (BJTC FAN993), YXY101 (BJTC FAN994), YXY102 (BJTC FAN1002), in soil under *Quercus* sp., 36°32.50′ N, 111°12.17′ E, alt. 1700 m, 26 October 2017, XYY070 (BJTC FAN1070); Huozhou country, Qilishangu, 36°35.44′ N, 112°1.46′ E, alt. 1800 m, 7 October 2020, in soil under *Quercus* sp., LT151 (BJTC FAN1266); Yuncheng City, Xia country, 35°4.44′ N, 111°23.41′ E, alt. 990 m, 7 October 2020, in soil under *Quercus* sp., LT152 (BJTC FAN1267), LT153 (BJTC FAN1268).

Notes: *Hymenogaster papilliformis* is characterized by the whitish to cream-colored basidiomata, and broadly fusiform to citriform basidiospores with a pronounced apex (2–3 μm, occasionally up to 4 μm long), distinct warts and ridges, and pronounced truncate appendix (2–3 μm long). DNA analysis showed that *H. papilliformis* shared less than 96.74% identity in the ITS sequence to other *Hymenogaster* species. The sequences of *H. papilliformis* clustered together on an independent branch in the ITS/LSU-based phylogenetic tree (Figure 1) with high supports, further supporting the erection of this new species. 

*Hymenogaster perisporius* T. Li & L. Fan, sp. nov. (Figure 6)

MycoBank: MB852602

Etymology: *perisporius*, referring to the gelatinous perisporium of basidiospores.

Holotype: China. Beijing, Mentougou County, Qingshui Town, Baihuashan Nature Reserve, alt. 738 m, 13 October 2016, in soil under *Populus beijingensis* W. Y. Hsu, WYW021 (BJTC FAN651).

Description—the basidiome is subglobose to irregular globose, 0.5–2 cm diameter, soft and elastic, dirty white to pale yellow when fresh, sometimes brownish, with a distinct depression at the white sterile base. Surface smooth, glabrous. 

The peridium is 223–310 μm thick, pseudoparenchymatous, composed of elliptic cells of 8–12 μm in diameter and interwoven hyphae of 6.8–8.5 μm broad, light yellowish to nearly hyaline. The gleba are reddish-brown to brown when fresh, deep reddish-brown when dry, loculate, with locules irregular, empty, and filled with spores at maturity. The hymenium is 21–29 μm thick. The hymenial cystidia are clavate, 26–36 µm long, only present when young, collapsed, and disappeared at maturity. The basidia clavate are not inflate on the apex, two-to-three-spored, mostly two-spored, with sterigmata short, 1–2 μm long, and basidia collapsed and disappeared at maturity. The basidiospores are ellipsoidal, yellow-brown to dark brown at maturity, ornamented with warts and ridges of 1–2 μm high, (15–)17–22(–23) × (11–)12–15(–17) μm (L_m_ × W_m_ = 19.1 ± 1.3 × 13.3 ± 0.9, *n* = 30), Q = 1.3–1.6 (Q_av_ = 1.44), excluding ornamentations, with gelatinous perisporium, with apex, obtuse, nearly hyaline, 1–2 μm high, with appendix, truncate, (1–)2–3 μm long.

Habit and habitat: hypogeous, gregarious, in the soil under Betula platyphylla, Castanea mollissima, Larix principis-rupprechtii Mayr., Pinus armandii Franch., P. tabuliformis Carrière, Populus beijingensis, Quercus liaotunggensis Koidz, and Q. mongolica Fisch. ex Ledeb.

Additional specimens examined. China. Beijing, Mentougou County, Qingshui Town, Baihuashan Nature Reserve, alt. 738 m, 4 August 2016, in soil under *Quercus mongolica*, HBD017 (BJTC FAN546), alt. 752 m, in soil under *Populus beijingensis*, X.Y. Yan 025 (BJTC FAN650), Y.W. Wang 022 (BJTC FAN653), Y.W. Wang 023 (BJTC FAN654). Shanxi Province: Yuncheng City, Yuanqu County, Lishan Town, Shunwangping scenic spot, alt. 2209 m, 16 August 2016, in soil under *Pinus armandii*, K.B. Huang 011 (BJTC FAN553), K.B. Huang 001 (BJTC FAN561), X.Y. Yan 014 (BJTC FAN569), Y.W. Wang 007 (BJTC FAN570), X.Y. Sang 004 (BJTC FAN589), B.D. He 002 (BJTC FAN592), B.D. He 003 (BJTC FAN6046), K.B. Huang 015 (BJTC FAN608), alt. 2276 m, 17 October 2016, K.B. Huang 041 (BJTC FAN694); Xinzhou City, Qiuqiangou, alt. 2099 m, 17 October 2016, in soil under *Larix principis-rupprechtii*, K.B. Huang 063 (BJTC FAN768); Lvliang City, Jiaocheng County, Shenweigou, alt. 2003 m, 7 September 2017, in soil under *Betula platyphylla*, X.Y. Yan 059 (BJTC FAN846), in soil under *Larix principis-rupprechtii*, X.Y. Yan 063 (BJTC FAN850); Linfen City, Pu Country, Chaoyanggou, alt. 1660 m, 11 September 2017, in soil under *Pinus tabuliformis*, K.B. Huang 125 (BJTC FAN952), X.Y. Yan 086 (BJTC FAN977); Guancen Mountain, Liaowangtai, Yingbeimian, alt. 2120 m, 13 October 2017, in soil under *Betula platyphylla*, K.B. Huang 139 (BJTC FAN1038), X.Y. Yan 118 (BJTC FAN1049); Linfen City, Pu County, Wulu Mountain, alt. 1555 m, 26 October 2017, in soil under *Quercus liaotunggensis*, K.B. Huang 143 (BJTC FAN1076); Yuncheng City, Xia County, Sijiao Town, Jialu Village, alt. 1057 m, 29 October 2017, in soil under *Castanea mollissima*, K.B. Huang 162 (BJTC FAN1126). 

Notes: *Hymenogaster perisporius* is characterized by the dirty white to pale yellow basidiomes, broad ellipsoidal to ellipsoidal, yellow-brown to dark brown basidiospores with warts and ridges, with gelatinous perisporium. *Hymenogaster bulliardii* Vittad. and *H. thwaitesii* Berk. and Broome are similar to *H. perisporius* in spore size and gleba color, but *H. bulliardii* has smooth spores and *H. thwaitesii* has a yellow-brown peridium. *Hymenogaster citrinus* is similar to *H. perisporius* in the appearance of its basidiospores, but the spores of *H. citrinus* are larger (21.1–25.9 × 14.0–18.4 μm). DNA analysis showed that sequences of *H. perisporius* clustered with *H. griseus* s. l. [2] (Figure 1); however, the spores are slender fusiform (Q = 1.9) in the latter [2], quite different from this new species. The ITS-LSU-based phylogeny supports the erection of this new species. The DNA analysis showed that *H. perisporius* shared less than 97% identity in ITS sequence to other *Hymenogaster* species.

*Hymenogaster variabilis* L. Fan & T. Li, sp. nov. (Figure 7)

MycoBank: MB852603

Etymology: *variabilis*, referring to the peridium of very variable thickness.

Holotype: China. Beijing, Mentougou County, Qingshui Town, Baihuashan Nature Reserve, alt. 1593 m, 14 October 2016, in soil under *Pinus tabulaeformis*, WYW024 (BJTC FAN656). 

Description—the basidiome is subglobose to irregular globose, 1–1.8 cm diameter, soft and elastic, pale yellow to earth yellow when fresh, yellow-brown to brown when dry, with a distinct depression at the sterile base. The surface is smooth and glabrous. 

The peridium is 167–351 μm thick, pseudoparenchymatous, composed of elliptic to subglobose cells of 8–19 μm in diameter, light yellowish to nearly hyaline. The gleba are light yellow-brown to brown when fresh, loculate, with locules irregular, empty, and filled with spores at maturity. The hymenium are 14–23 μm thick. The hymenial cystidia are clavate, 25–40 µm long, only present when young, collapsed, and disappeared at maturity. The basidia are narrow clavate, occasionally inflate at the apex, two-spored, 27–36 μm, sterigmata short, 1–3 μm long, basidia collapsed and disappeared at maturity. The basidiospores are broadly ellipsoidal to subglobose, pale yellow-brown to yellow-brown at maturity, ornamented with sparse warts and ridges of 1–1.2 μm high, with ridges short, interwoven, (8.5–)10.6–13.6(–15) × (7.4–)8.8–10.9(–13) μm (L_m_ × W_m_ = 11.5 ± 1.7 × 9.7 ± 1.4, *n* = 30), Q = 1.1–1.3 (Q_av_ = 1.2), excluding ornamentations, without gelatinous perisporium, with apex, obtuse, papillary, nearly hyaline, 1–3 μm high, with appendix, truncate, 1–2 μm long.

Habit and habitat: hypogeous, gregarious, in the soil under *Pinus tabulaeformis*.

Additional specimen examined. China. Shanxi Province, Yuncheng City, Yuanqu County, Lishan Town, Shunwangping, alt. 1744 m, 30 October 2017, LT071 (BJTC FAN1141).

Notes: *Hymenogaster variabilis* is characterized by the peridium with significant changes in thickness, broad ellipsoidal to subglobose basidiospores ornamented with sparse warts and ridges. *Hymenogaster minisporus* is similar to *H. variabilis* in spore size, but the former is distinguished by its densely ornamented spores, and its peridium of 110–288 μm thickness. The ITS-LSU-based phylogeny supports the erection of this new species. The DNA analysis showed that *H. variabilis* shared less than 95.65% identity in the ITS sequence to other *Hymenogaster* species.

## 4. Discussion

Among the 32 species of *Hymenogaster* reported in previous studies from China, 14 species were recorded from Shanxi Province [19], but none from Beijing. We re-examined those voucher specimens collected at that time and confirmed that morphologically they represented species of *Hymenogaster*, but unfortunately their DNA sequences could not be successfully sequenced. Among our newly collected specimens from Shanxi Province, two species reported in the previous studies were confirmed with molecular data in this study, namely, *H. arenarius* and *H. citrinus* (Figure 1 and Figure 2). DNA analysis showed that the two species shared less than 95.43% and 95.26% sequence identity, respectively, with other *Hymenogaster* species. Consequently, there are a total of seven species currently confirmed using morphological and molecular data in Shanxi and Beijing, including five new species described in this study. A key to these species is provided as follows. 

Key to the species of *Hymenogaster* from Shanxi and Beijing in China:
1Basidiome pale yellow, white to dirty white when fresh21Basidiome earth yellow to yellow-brown when fresh32Gleba reddish brown to brown when fresh42Gleba light brown when fresh*H. minisporus*3Peridium very variable in thickness and more than >130 μm thick53Peridium without major changes in thickness and <130 μm thick64Basidiospores 21–26 × 14–18.5 μm*H. citrinus*4Basidiospores 17–22 × 12–15 μm*H. perisporius*5Basidiospores broad ellipsoidal to subglobose, Q = 1.1–1.3*H. variabilis*5Basidiospores broad fusiform to citriform, Q = 1.3–1.4*H. apilliformis*6Basidiospores fusiform, Q = 1.2–1.4*H. arenarius*6Basidiospores broadly ellipsoidal to broadly ovoid, Q = 1.1–1.3*H. latisporus*

## Figures and Tables

**Figure 1 jof-10-00272-f001:**
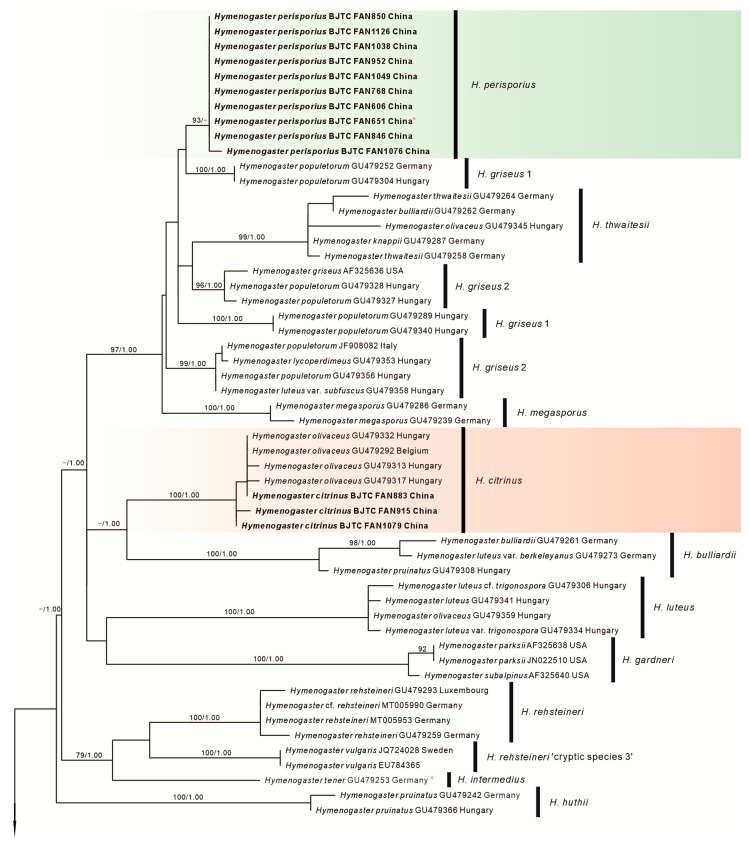
Phylogeny derived from maximum likelihood analyses of the ITS/LSU sequences from *Hymenogaster* and related species. Two sequences of *Anamika lactariolens* were selected as the outgroup. Values on the left represent the likelihood bootstrap support values (≥70%). Values on the right represent significant Bayesian posterior probability values (≥0.95). Novel sequences are in bold. Super index “H” means “Holotype”. The green background represents the five new species described in this study, the red background represents the two old species from China supported by molecular data.

**Figure 2 jof-10-00272-f002:**
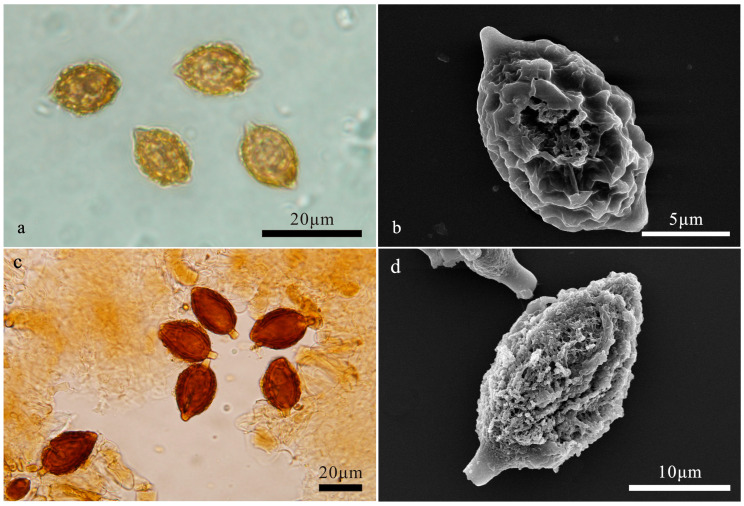
(**a**,**b**) *Hymenogaster arenarius*. (**a**) Basidiospores under LM. (**b**) Basidiospore under SEM. (**c**,**d**) *Hymenogaster citrinus*. (**c**) Basidiospores under LM. (**d**) Basidiospore under SEM.

**Figure 3 jof-10-00272-f003:**
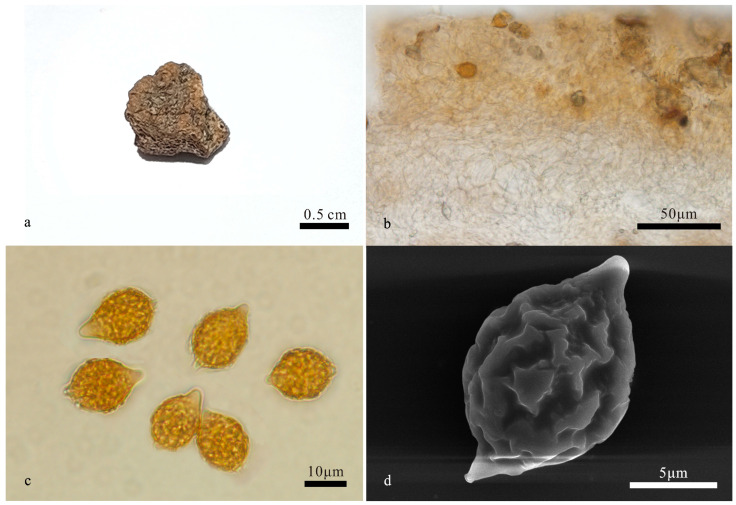
*Hymenogaster latisporus* (BJTC FAN1134, holotype). (**a**) Basidiome. (**b**) Peridium under LM. (**c**) Basidiospores under LM. (**d**) Basidiospore under SEM.

**Figure 4 jof-10-00272-f004:**
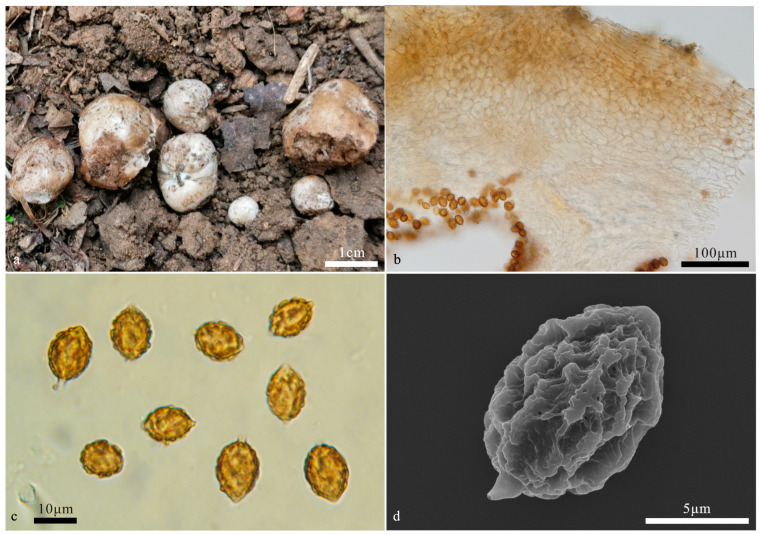
*Hymenogaster minisporus* (BJTC FAN1244, holotype). (**a**) Basidiomes. (**b**) Peridium under LM. (**c**) Basidiospores under LM. (**d**) Basidiospores under SEM.

**Figure 5 jof-10-00272-f005:**
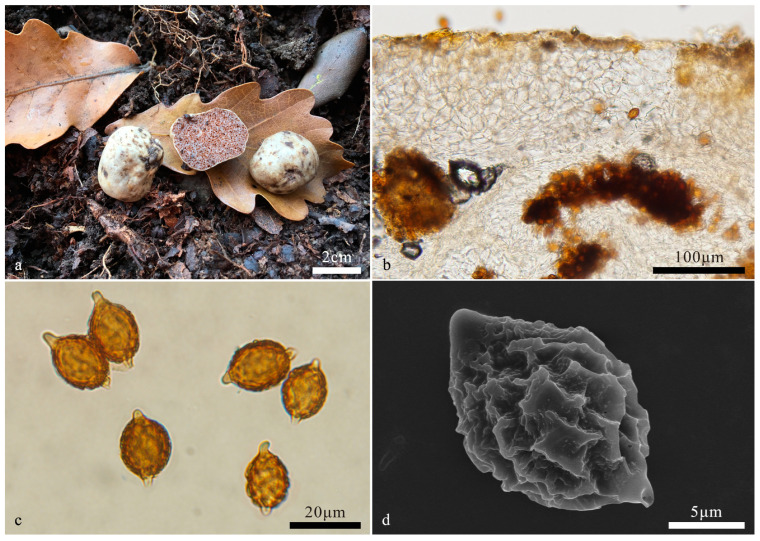
*Hymenogaster papilliformis* (BJTC FAN1074, holotype). (**a**) Basidiomes. (**b**) Peridium under LM. (**c**) Basidiospores under LM. (**d**) Basidiospores under SEM.

**Figure 6 jof-10-00272-f006:**
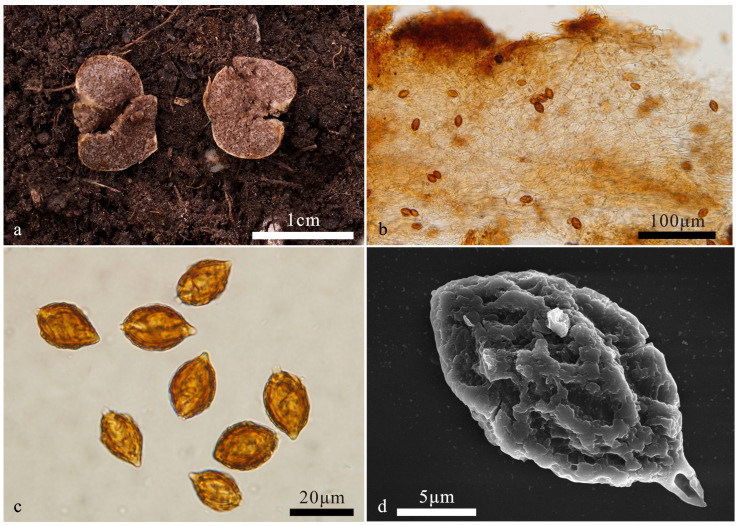
*Hymenogaster perisporius* (BJTC FAN651, holotype). (**a**) Basidiomes. (**b**) Peridium under LM. (**c**) Basidiospores under LM. (**d**) Basidiospores under SEM.

**Figure 7 jof-10-00272-f007:**
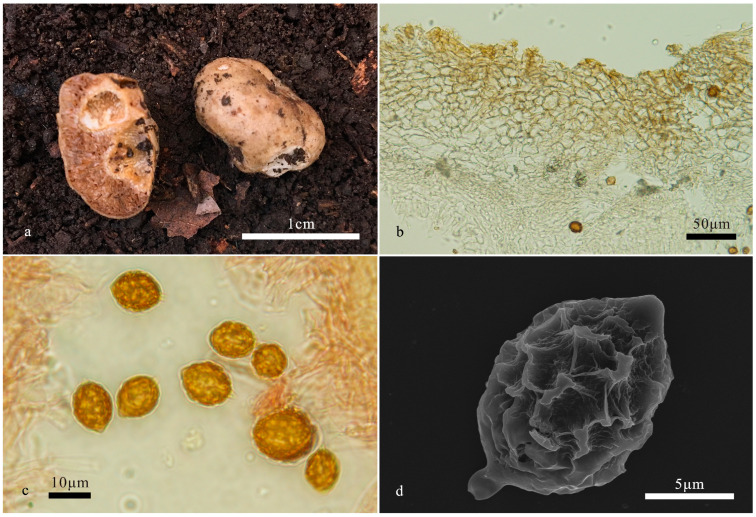
*Hymenogaster variabilis* (BJTC FAN656, holotype). (**a**) Basidiomes. (**b**) Peridium under LM. (**c**) Basidiospores under LM. (**d**) Basidiospores under SEM.

**Table 1 jof-10-00272-t001:** Sources of specimens and GenBank accession numbers for sequences used in this study. Newly generated sequences are in bold.

Taxon Name in Analysis	Taxon Name	Collection	Country	GenBank Accession Number	
				ITS	nrLSU
*Anamika lactariolens* AY818352	*Anamika lactariolens*	taxon:301353		AY818352	-
*Anamika lactariolens* NR_119524	*Anamika lactariolens*	HC 88/95		NR_119524	-
** *Hymenogaster arenarius* ** **BJTC FAN786 China**	** *Hymenogaster arenarius* **	**BJTC FAN786**	**China**	**PP467413**	**PP467449**
** *Hymenogaster arenarius* ** **BJTC FAN856 China**	** *Hymenogaster arenarius* **	**BJTC FAN856**	**China**	**PP467414**	**PP467450**
*Hymenogaster arenarius* GU479233 Germany	*Hymenogaster arenarius*	it10_26_2	Germany	GU479233	-
*Hymenogaster arenarius* GU479272 Germany	*Hymenogaster arenarius*	it5_2	Germany	GU479272	-
*Hymenogaster arenarius* GU479278 Germany	*Hymenogaster arenarius*	it6_3	Germany	GU479278	-
*Hymenogaster bulliardii* GU479261 Germany	*Hymenogaster bulliardii*	it20_4	Germany	GU479261	-
*Hymenogaster bulliardii* GU479262 Germany	*Hymenogaster thwaitesii*	it20_4_1	Germany	GU479262	-
*Hymenogaster* cf. *niveus* MT005942 Germany	*Hymenogaster* xxx	KR-M-0044217	Germany	MT005942	-
*Hymenogaster* cf. *niveus* MT005967 Germany	*Hymenogaster niveus* ‘cryptic species 3’	KR-M-0044314	Germany	MT005967	-
*Hymenogaster* cf. *rehsteineri* MT005990 Germany	*Hymenogaster rehsteineri*	KR-M-0044423	Germany	MT005990	-
** *Hymenogaster citrinus* ** **BJTC FAN1079 China**	** *Hymenogaster citrinus* **	**BJTC FAN1079**	**China**	**PP467412**	**PP467448**
** *Hymenogaster citrinus* ** **BJTC FAN883 China**	** *Hymenogaster citrinus* **	**BJTC FAN883**	**China**	**PP467410**	**PP467446**
** *Hymenogaster citrinus* ** **BJTC FAN915 China**	** *Hymenogaster citrinus* **	**BJTC FAN915**	**China**	**PP467411**	**PP467447**
*Hymenogaster glacialis* AF325634	*Hymenogaster* sp.	GP 5302	-	AF325634	-
*Hymenogaster griseus* AF325636 USA	*Hymenogaster griseus* 2	Trappe 12841	USA	AF325636	-
*Hymenogaster knappii* GU479287 Germany	*Hymenogaster thwaitesii*	it9_2	Germany	GU479287	-
** *Hymenogaster latisporus* ** **BJTC FAN1134 China**	** *Hymenogaster latisporus* **	**BJTC FAN1134, holotype**	**China**	**PP467404**	**PP467440**
*Hymenogaster luteus* cf. *trigonospora* GU479306 Hungary	*Hymenogaster luteus*	zb1457	Hungary	GU479306	-
*Hymenogaster luteus* GU479341 Hungary	*Hymenogaster luteus*	zb2603	Hungary	GU479341	-
*Hymenogaster luteus* var. *berkeleyanus* GU479273 Germany	*Hymenogaster bulliardii*	it5_21	Germany	GU479273	-
*Hymenogaster luteus* var. *subfuscus* GU479358 Hungary	*Hymenogaster griseus* 2	zb37	Hungary	GU479358	-
*Hymenogaster luteus* var. *trigonospora* GU479334 Hungary	*Hymenogaster luteus*	zb235	Hungary	GU479334	-
*Hymenogaster lycoperdineus* GU479353 Hungary	*Hymenogaster griseus* 2	zb3533	Hungary	GU479353	-
*Hymenogaster megasporus* GU479239 Germany	*Hymenogaster megasporus*	it12_1	Germany	GU479239	-
*Hymenogaster megasporus* GU479286 Germany	*Hymenogaster megasporus*	it8_5_1	Germany	GU479286	-
** *Hymenogaster minisporus* ** **BJTC FAN1244 China**	** *Hymenogaster minisporus* **	**BJTC FAN1244, holotype**	**China**	**PP467407**	**PP467443**
*Hymenogaster niveus* GU479255 Germany	*Hymenogaster niveus* ‘cryptic species 1’	it17_3	Germany	GU479255	-
*Hymenogaster niveus* GU479307 Hungary	*Hymenogaster* xxx	zb1461	Hungary	GU479307	-
*Hymenogaster niveus* GU479344 Hungary	*Hymenogaster niveus* ‘cryptic species 3’	zb28	Hungary	GU479344	-
*Hymenogaster niveus* KU878613 USA	*Hymenogaster* xxx	SC14_3	USA	KU878613	-
*Hymenogaster olivaceus* GU479292 Belgium	*Hymenogaster citrinus*	dt8293	Belgium	GU479292	-
*Hymenogaster olivaceus* GU479313 Hungary	*Hymenogaster citrinus*	zb1645	Hungary	GU479313	-
*Hymenogaster olivaceus* GU479317 Hungary	*Hymenogaster citrinus*	zb1817	Hungary	GU479317	-
*Hymenogaster olivaceus* GU479332 Hungary	*Hymenogaster citrinus*	zb2300	Hungary	GU479332	-
*Hymenogaster olivaceus* GU479345 Hungary	*Hymenogaster thwaitesii*	zb2804	Hungary	GU479345	-
*Hymenogaster olivaceus* GU479359 Hungary	*Hymenogaster luteus*	zb3721	Hungary	GU479359	-
** *Hymenogaster papilliformis* ** **BJTC FAN1002 China**	** *Hymenogaster papilliformis* **	**BJTC FAN1002**	**China**	**PP467396**	**PP467432**
** *Hymenogaster papilliformis* ** **BJTC FAN1070 China**	** *Hymenogaster papilliformis* **	**BJTC FAN1070**	**China**	**PP467399**	**PP467435**
** *Hymenogaster papilliformis* ** **BJTC FAN1074 China**	** *Hymenogaster papilliformis* **	**BJTC FAN1074, holotype**	**China**	**PP467400**	**PP467436**
** *Hymenogaster papilliformis* ** **BJTC FAN1109 China**	** *Hymenogaster papilliformis* **	**BJTC FAN1109**	**China**	**PP467402**	**PP467438**
** *Hymenogaster papilliformis* ** **BJTC FAN1156 China**	** *Hymenogaster papilliformis* **	**BJTC FAN1156**	**China**	**PP467406**	**PP467442**
** *Hymenogaster papilliformis* ** **BJTC FAN1266 China**	** *Hymenogaster papilliformis* **	**BJTC FAN1266**	**China**	**PP467408**	**PP467444**
** *Hymenogaster papilliformis* ** **BJTC FAN1267 China**	** *Hymenogaster papilliformis* **	**BJTC FAN1267**	**China**	**PP467409**	**PP467445**
** *Hymenogaster papilliformis* ** **BJTC FAN655 China**	** *Hymenogaster papilliformis* **	**BJTC FAN655**	**China**	**PP467381**	**PP467417**
** *Hymenogaster papilliformis* ** **BJTC FAN807 China**	** *Hymenogaster papilliformis* **	**BJTC FAN807**	**China**	**PP467384**	**PP467420**
** *Hymenogaster papilliformis* ** **BJTC FAN820 China**	** *Hymenogaster papilliformis* **	**BJTC FAN820**	**China**	**PP467385**	**PP467421**
** *Hymenogaster papilliformis* ** **BJTC FAN891 China**	** *Hymenogaster papilliformis* **	**BJTC FAN891**	**China**	**PP467388**	**PP467424**
** *Hymenogaster papilliformis* ** **BJTC FAN944 China**	** *Hymenogaster papilliformis* **	**BJTC FAN944**	**China**	**PP467389**	**PP467425**
** *Hymenogaster papilliformis* ** **BJTC FAN958 China**	** *Hymenogaster papilliformis* **	**BJTC FAN958**	**China**	**PP467391**	**PP467427**
** *Hymenogaster papilliformis* ** **BJTC FAN960 China**	** *Hymenogaster papilliformis* **	**BJTC FAN960**	**China**	**PP467392**	**PP467428**
** *Hymenogaster papilliformis* ** **BJTC FAN980 China**	** *Hymenogaster papilliformis* **	**BJTC FAN980**	**China**	**PP467393**	**PP467429**
** *Hymenogaster papilliformis* ** **BJTC FAN983 China**	** *Hymenogaster papilliformis* **	**BJTC FAN983**	**China**	**PP467394**	**PP467430**
** *Hymenogaster papilliformis* ** **BJTC FAN992 China**	** *Hymenogaster papilliformis* **	**BJTC FAN992**	**China**	**PP467395**	**PP467431**
*Hymenogaster parksii* AF325638 USA	*Hymenogaster gardneri*	Trappe 13296	USA	AF325638	-
*Hymenogaster parksii* JN022510 USA	*Hymenogaster gardneri*	SOC1643	USA	JN022510	-
** *Hymenogaster perisporius* ** **BJTC FAN1038 China**	** *Hymenogaster perisporius* **	**BJTC FAN1038**	**China**	**PP467397**	**PP467433**
** *Hymenogaster perisporius* ** **BJTC FAN1049 China**	** *Hymenogaster perisporius* **	**BJTC FAN1049**	**China**	**PP467398**	**PP467434**
** *Hymenogaster perisporius* ** **BJTC FAN1076 China**	** *Hymenogaster perisporius* **	**BJTC FAN1076**	**China**	**PP467401**	**PP467437**
** *Hymenogaster perisporius* ** **BJTC FAN1126 China**	** *Hymenogaster perisporius* **	**BJTC FAN1126**	**China**	**PP467403**	**PP467439**
** *Hymenogaster perisporius* ** **BJTC FAN606 China**	** *Hymenogaster perisporius* **	**BJTC FAN606**	**China**	**PP467379**	**PP467415**
** *Hymenogaster perisporius* ** **BJTC FAN651 China**	** *Hymenogaster perisporius* **	**BJTC FAN651, holotype**	**China**	**PP467380**	**PP467416**
** *Hymenogaster perisporius* ** **BJTC FAN768 China**	** *Hymenogaster perisporius* **	**BJTC FAN768**	**China**	**PP467383**	**PP467419**
** *Hymenogaster perisporius* ** **BJTC FAN846 China**	** *Hymenogaster perisporius* **	**BJTC FAN846**	**China**	**PP467386**	**PP467422**
** *Hymenogaster perisporius* ** **BJTC FAN850 China**	** *Hymenogaster perisporius* **	**BJTC FAN850**	**China**	**PP467387**	**PP467423**
** *Hymenogaster perisporius* ** **BJTC FAN952 China**	** *Hymenogaster perisporius* **	**BJTC FAN952**	**China**	**PP467390**	**PP467426**
*Hymenogaster populetorum* GU479252 Germany	*Hymenogaster griseus* 1	it16_1_1	Germany	GU479252	-
*Hymenogaster populetorum* GU479289 Hungary	*Hymenogaster griseus* 1	aszodvt_1991	Hungary	GU479289	-
*Hymenogaster populetorum* GU479304 Hungary	*Hymenogaster griseus* 1	zb1436	Hungary	GU479304	-
*Hymenogaster populetorum* GU479327 Hungary	*Hymenogaster griseus* 2	zb2097	Hungary	GU479327	-
*Hymenogaster populetorum* GU479328 Hungary	*Hymenogaster griseus* 2	zb2105	Hungary	GU479328	-
*Hymenogaster populetorum* GU479340 Hungary	*Hymenogaster griseus* 1	zb2576	Hungary	GU479340	-
*Hymenogaster populetorum* GU479356 Hungary	*Hymenogaster griseus* 2	zb3594	Hungary	GU479356	-
*Hymenogaster populetorum* JF908082 Italy	*Hymenogaster griseus* 2	17022	Italy	JF908082	-
*Hymenogaster pruinatus* GU479242 Germany	*Hymenogaster huthii*	it12_3_1	Germany	GU479242	-
*Hymenogaster pruinatus* GU479308 Hungary	*Hymenogaster bulliardii*	zb1485	Hungary	GU479308	-
*Hymenogaster pruinatus* GU479366 Hungary	*Hymenogaster huthii*	zb95	Hungary	GU479366	-
*Hymenogaster rehsteineri* GU479259 Germany	*Hymenogaster rehsteineri*	it2_4_1	Germany	GU479259	-
*Hymenogaster rehsteineri* GU479293 Luxembourg	*Hymenogaster rehsteineri*	dt8455	Luxembourg	GU479293	-
*Hymenogaster rehsteineri* MT005953 Germany	*Hymenogaster rehsteineri*	KR-M-0044018	Germany	MT005953	-
*Hymenogaster rubyensis* AY945303 USA	*Hymenogaster* sp.	Fogel 2698	USA	AY945303	-
*Hymenogaster* sp. MK027200 Slovenia	*Hymenogaster niveus* ‘cryptic species 1’	FV4_04	Slovenia	MK027200	-
*Hymenogaster subalpinus* AF325640 USA	*Hymenogaster gardneri*	Trappe 22752	USA	AF325640	-
*Hymenogaster tener* EU784363 UK	*Hymenogaster tener*	RBG Kew K(M)102406	UK	EU784363	-
*Hymenogaster tener* GU479250 Germany	*Hymenogaster tener*	it15_3	Germany	GU479250	-
*Hymenogaster tener* GU479253 Germany	*Hymenogaster intermedius*	it16_2, holotype	Germany	GU479253	-
*Hymenogaster thwaitesii* GU479258 Germany	*Hymenogaster thwaitesii*	it2_2	Germany	GU479258	-
*Hymenogaster thwaitesii* GU479264 Germany	*Hymenogaster thwaitesii*	it3_2	Germany	GU479264	-
** *Hymenogaster variabilis* ** **BJTC FAN1141 China**	** *Hymenogaster variabilis* **	**BJTC FAN1141**	**China**	**PP467405**	**PP467441**
** *Hymenogaster variabilis* ** **BJTC FAN656 China**	** *Hymenogaster variabilis* **	**BJTC FAN656, holotype**	**China**	**PP467382**	**PP467418**
*Hymenogaster vulgaris* EU784365	*Hymenogaster rehsteineri* ‘cryptic species 3’	RBG Kew K(M)27363	-	EU784365	-
*Hymenogaster vulgaris* JQ724028 Sweden	*Hymenogaster rehsteineri* ‘cryptic species 3’	GN_4d_I	Sweden	JQ724028	-
Uncultured Agaricales HM105539 China	*Hymenogaster minisporus*	QL054	China	HM105539	-
Uncultured fungus EU554705 Canada	*Hymenogaster* sp.	A2N_88	Canada	EU554705	-
Uncultured fungus EU554717 Canada	*Hymenogaster* sp.	A3E_60	Canada	EU554717	-
Uncultured *Hymenogaster* LT980461 China	*Hymenogaster minisporus*	taxon:522720	China	LT980461	-

## Data Availability

The sequencing data has been submitted to GenBank.

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
