# Peer review of "Five New Species of the Genus Hymenogaster (Hymenogastraceae, Agaricales) from Northern China"

_jof, 2024, doi:10.3390/jof10040272_

Round 1

Reviewer 1 Report

Major concerns:

The alignment was not uploaded to TreeBase or in another way provided to the reviewer, although requested from the editor, this is standard today in taxonomic work when such data is the base for segregation if taxa. Genbank numbers of the LSU data used in the analyses and the newly generated data is missing in Table 1.

- Language issues throughout the text

- Reference to some standard works on hypogeous fungi is missing, eg.

Montecchi & Sarasini 2000. Fungi ipogei d’Europa – contains descriptions and photos pf 16 Hymenogaster spp  + a dichotomous key

 Pegler, Spooner, Young 1995. British Truffles. Descriptions and key to 11 species of Hymenogaster

Species descriptions are informative and largely relevant although containing some terminology and language issues in need of adjustment and corrections:

Spore morphology: 

·      Basal sterigma – Montecchi & Sarasini  (2000) uses “sterigmal appendage”; Soehner (1962) :  “Appendix”

·      “Apical hump” – previous literature uses apex and papilla

·      Spore shape: what is the difference between “broad (sic!) ellipsoidal” and “broad (sic!) ovoid”?

Basidia 

·      ….xx-xx µm “over than hymenium” – not idiomatic English

Cystidia

·      Presence of cystidia is mentioned in some of the species; this needs to be clarified and prooved, as cystidia are not known to occur in Hymenogaster (cfr Soehner 1962: 9; “Cystiden kommen bei den Hymenogaster-Arten nicht vor”)

Reviewer 2 Report

It is an interesting manuscript with a very good contribution to the known of the genus Hymenogaster, with the description of new species and the confirmation of previous ones recorded in the country. Although the new species are based on morphological and molecular evidence, in some cases they are based on a single specimen. Unless a very good explanation is provided (e.g. the difficulty, in some cases not all, of finding more specimens), I do not agree that new species are based on a unique specimen.

I strongly recommend changing the peridium photos of some species or in the worst case scenary, not including them as they are.

I attach the pdf with my observations. It is important to say that I tried to indicate all of them, but I am sure that some were missing. It is the authors' task to find everything and correct the manuscript in all cases.

I made some suggestions related to improving English writing, but I'm not a native English speaker.
